# Isolation and identification of microorganisms associated with automated teller machines on Federal Polytechnic Ede campus

O. G. Dawodu *, R. B. Akanbi

Department of Science Laboratory Technology, Federal Polytechnic Ede, Ede, Osun State, Nigeria

* dawgrace@yahoo.com, dawodu.olufunke@federalpolyede.edu.ng

**Data Availability Statement:** All relevant data are contained within the manuscript and its supporting information files.

## Abstract

Automated Teller Machines (ATM) are visited everyday by millions of people. This machine is accessible to the general public irrespective of class, age or race. The contact point of all ATM machines is the hand which on their own are 'vaults' of microorganisms. An elaborate survey was taken for complete assessment of possible microbial contamination in the Federal Polytechnic Ede campus. Selected ATM machines on campus were used as case study to characterize, identify and determine the degree of bacterial contamination of microorganisms and their potential as reservoir of microbes. Swabs were collected from each ATM screen, buttons, floor, user's hand, and exposure of plates. After collection of the samples, they were plated in nutrient agar. The results showed the presence of increased bacterial count subsequently, most pathogens on characterization revealed the genus of the particular organisms *E. coli*, *Pseudomonas*, *Staphylococcus aureus*, *Klebsiella*, *Micrococcus*, *Salmonella* and *Serratia*. The study showed the potential hazard inherent in ATM machine usage and draws attention to our level of hand hygiene compliance.

## Introduction

Microorganisms are very small organisms, which can only be seen with the aid of microscope [1], they however, have both positive and negative usage roles. Microorganisms include bacteria, fungi and protoctists [2]. There are approximately one hundred and fifty nine thousand (159,000) species of microorganisms known to date, although this is thought to be less than five percent (5%) of the total microbes in existence [3]. Microorganisms are ubiquitous and have an amazing ability to adapt to new environments and further multiply in large numbers within a limited time [4]. Their ability to adapt and multiply on various surfaces and in different environments is key to their being found on soil surfaces, acidic hot springs radioactive waste water, deep in the earth's crust as well as organic matter and life bodies of flora and fauna [5]. With this interesting fact in mind, the ready familiarity of microbes with hardware interfaces such as cyber appliances and its users calls for carrying out experimental studies to show the linkages between the three.

The United State Centre for disease control (USCDC) in 2005 found out, that microbes could find exchange between contaminated hands and cyber appliances such as the surface of

**Funding:** The financial disclosure of the manuscript is for this study is that 'no financial or material support was received for this study as no funding organization or body was involved in the study design, data collection and analysis, decision to publish, or preparation of the manuscript." We the authors state specifically that 'we received no specific funding for this work'. Our institution Federal Polytechnic Ede, was not in any way financially involved with this study, as no funding in anyway was received from them. This study was clearly an independent research work with no affiliations to any grant awarding organization whatsoever.

**Competing interests:** The authors have declared that no competing interests exist.

automated teller machines. The ATM in question is a telecommunication device that aids the customers of banks to carry out transactions with ease, irrespective of time or place [6, 7].

The use of hardware interfaces such as the keyboard, mouse and ATM keypad has greatly expanded over the past few years with the development of various forms of computer-based management applications. With the advent of modernization, the personal computers has found application in every sphere of the globe due to their relatively low price and their ease of usage solely made possible by the advent of the graphical interface (GUI) [8]. In today's world, the advent and use of computer systems and consequently interfaces is fast rising in schools, offices, cybercafés and hospitals, so that interfaces have found wide application in almost every occupational, recreational and residential environment [9]. This wild growth has consequently led to regular and unrestricted sharing of interfaces among users. With the ease at which microorganisms acquired from the human microflora or as transient organisms from the environment, and previous accounts of cross contamination of microorganisms [10, 11], it is readily seen that pathogens could be transferred among users who share interfaces [12].

The automated teller machine (ATM) machine is regarded as a mini bank as almost all forms of bank transactions can be carried out on it [7]. Its key pads on which pathogenic microorganisms might survive, represent an often overlooked reservoir for enteric diseases [13]. A representative amount of microbes bear the potentials for survival on dry fomites like ATM machine key pads. They have evolved different physiological resting stages, which gives them the advantage for surviving or hibernating due to low water activity. Some gram-negative bacteria can remain as long as eleven days on surfaces [14].

Important factors for the survival of pathogenic on surfaces are the presence of organic matter, solar irradiation, temperature and humidity [15]. A review reported that many Gram-positive bacteria such as *Enterococcus spp.*, *Staphylococcus aureus* and *Streptococcus pyogenes* and Gram-negative bacteria such as *Acinetobacter spp.*, *Escherichia coli*, *Klebsiella spp.*, *Pseudomonas aeruginosa* and *Shigella spp.* can survive for months on surfaces [16, 17]. The survival rate of most pathogens vary with *mycobacteria* and *Clostridium difficile* having monthly lifespan, whereas *Bordetellapertussis*, *Haemophilus influenza* and *Vibrio cholera* persist only for days [18, 19].

Specific bacteria such as *Salmonella* and *Escherichia coli* have been implicated to be transferred from the hand to raw processed and cooked foods, even at minute levels on the fingers [7, 20–22]. In 2002, Kissiedu's study [23] showed that snacks eaten with the fingers can easily be cross contaminated by bacteria from the hands through constant exchange of dirty currency notes [24]. It has also been reported that fungal microbial contamination is also found associated with use of ATMs [25]. Some studies have found out that microbes once in contact with hand and some hard surfaces find easy habitat with such surfaces and as a result are quite difficult to get rid of [26, 27].

A lot of studies have been done on microbial contamination on ATMs [7, 9, 12, 25, 28–30], most of them citing the dangers inherent in frequent usage of ATMs, but to date nothing tangible has been done to reduce such contaminations, with people using the ATMs without the slightest bother of hygiene compliance, while most people after using the rest rooms are conscious of washing their hands, the same cannot be said for usage of ATM machines. The banks themselves are somewhat guilty of this, apart from circulating old currencies, there are almost no preventive measures attached to these devices. With the advent of the recent COVID 19 pandemic and so many other futuristic pandemic with microbial transmission format, it is necessary for everyone to be aware of the imminent danger of such devices and enforce reduction in contamination of such devices.

This aim of this study is to isolate, identify and determine the degree of bacterial contamination of Automated Teller machines (ATM) and possible health implications of such on

Federal Polytechnic Ede campus, Ede Osun state. It has been observed that microbial contaminations are limitless especially in developing countries where most users of automated teller machines (ATM) are largely ignorant of the potential hazards they face each time they use an automated teller machine (ATM).

## Materials and methods

### Study area

This study was carried out in Federal Polytechnic Ede, Osun State, Nigeria. A particular bank (Access Bank) was used for the sample collection. The reason for this was that it had 2 ATM machines dispensing 1000 and 500 naira notes. Different times were used for the collection; early hours of the morning and peak afternoon periods. This bank was selected based on the fact that it had 2 ATM machines, dispensed different currencies and was the most visited ATM machine on Campus at the time of this study.

### Sample collection and processing

A total of twenty eight (28) swab samples were collected from the Bank's automated teller machines located on Federal Polytechnic Ede campus, Ede, Osun state with the aid of sterile cotton swab sticks moistened with sterile distilled water before swabbing the buttons of the ATM machines. The swab sticks were then transferred to the laboratory within two hours of collection for bacteriological analysis. This bank's ATM machines themselves are actually situated in close proximity to the lab where the study was carried out.

**Bacterial inoculation.** The inoculation of bacteria involved the direct streaking of the swab sticks on nutrient agar in petri dishes each labelled according to date and source of sample code. The streaked sample was incubated for 24 hours at 37°C after which the colonies were observed.

### Materials

The materials used include glass wares such as MacCartney bottles, beaker, conical flasks, measuring cylinder, glass slides, inoculating wire loop, aluminium foil, cotton wool, swab sticks and spirit lamp.

### Media/Agar

Nutrient and citrate agar.

### Reagents

Distilled water, methylated spirit, ethanol and stains such as safranin, crystal violet and Gram iodine.

### Washing and sterilizing of materials

All glass wares were washed with detergent and were air dried. The glass wares were packed with aluminium foil into canisters and were placed into hot air oven for sterilization at 16°C for 2 hours. The wares were brought out of the oven and were allowed to cool after sterility has been achieved. They were kept for storage when needed. Work surfaces were cleaned and sterilized by swabbing with 95% ethanol. Aseptic working environment was achieved with the use of spirit lamp.

### Preparation of media

The media used (NA and citrate agar) were weighed and prepared according to manufacturer's specification. The prepared media was carefully packed into the autoclave and sterilized at 121˚C for 15 minutes. Prior to use, the media were cooled to about 45˚C.

### Isolation and identification of bacterial isolates

Plate growths were noticed after 24 hours incubation, the isolates were then sub-cultured on fresh media plates until pure isolates were observed. The pure culture of isolates were stocked into MacCartney bottles. The isolates were identified based on their morphological appearance [31], Gram reaction and biochemical characteristics.

### Gram staining techniques

A thin smear was made by emulsifying a little portion of organisms picked from stocked colony of 18–24 hours old pure culture into a drop of sterile distilled water on a grease free slide. The smear was air dried and heat fixed by passing it slightly over flame. The slide was carefully placed on the staining rack and was flooded with primary stain (crystal violet) for 30–60 seconds. Gram's iodine was added (mordant) for 30 seconds. The smear was gently rinsed with tap water. 70% ethanol was applied as decolouriser for 10–30 seconds; it was the stained with the secondary stain (safranin) for 30 seconds before rinsing with tap water and was allowed to dry. The smear was examined under the microscope using oil immersion objective (x100). Gram positive organisms appeared purple while Gram negative appeared red.

### Biochemical characterization of the isolates

These tests were carried out to further identify and classify the isolates. They include; Catalase test [32], coagulase test [33], this test is used to differentiate *Staphylococcus aureus* (positive) from coagulase negative *Staphylococci*, oxidase test [34], Citrate utilization test [35], motility test [35], indole Test [36], urea hydrolysis (urease test) [34], sugar fermentation test (glucose, sucrose, lactose, galactose, maltose and fructose) [37] respectively.

### Oxidase test

This test is used to identify microorganisms containing the enzyme cytochrome oxidase (important in the electron transport chain). It is commonly used to distinguish between oxidase negative *Enterobacteriaceae* and oxidase positive *Pseudomadaceae*.

A piece of filter paper was soaked with a few drops of oxidase reagent (*Tetramethyl-p-phenylenediaminedihydrochloride*). A colony of the test organism was then smeared on the soaked filter paper. If the organism could produce oxidase, the phenylenediamine in the reagent will be oxidized to deep purple color. The change of color within 10 seconds indicates positive result.

### Sugar fermentation test

The carbohydrate fermentation test is used to determine whether or not bacteria can ferment a specific carbohydrate. Carbohydrate fermentation patterns are useful in differentiating among bacterial groups or species.

It tests for the presence of acid and/or gas produced from carbohydrate fermentation. Basal medium containing a single carbohydrate source such as glucose, lactose, sucrose or any other carbohydrate is used for this purpose. A pH indicator bromothymol blue (BTB), is also present in the medium; which will detect the lowering of the pH of the medium due to acid

production. Small inverted tubes called Durham tube is also immersed in the medium to test for the production of the gas (hydrogen or carbon dioxide). It's a positive test for all members of Enterobacteriaceae.

### Catalase test

This test is used to identify organisms that produce the enzyme catalase. This enzyme detoxifies hydrogen peroxide ($H_2O_2$) by breaking it down into water and oxygen gas. This test demonstrates the presence of catalase, an enzyme characterized with the release of oxygen from hydrogen peroxide. A drop of 3% hydrogen peroxide solution was added to the sterile slide containing a loopful of the organism. Foaming or bubble indicates a positive result.

### Indole test

This test is used to identify microbes that can break down tryptophan to indole. It is used to identify bacteria of the family Enterobacteriaceae. Innocuate sterilized tubes containing tryptophan broth (4 ml) and incubate tubes for 24–28 hrs. After which 0.5 ml of Kovac's reagent is added. Presence/absence of ring indicates positive/negative test.

### Citrate utilization test

This test is often used to differentiate organisms that are capable of utilizing citrate as a carbon source. Simmon's citrate agar medium was prepared in bijou bottle and allowed to set in a slanting position. A sterile wire loop was used to inoculate the test organism on to the slant medium and incubated at 37˚C for 48 hours after which it was examined for color change. A bright blue color in the medium gave a positive citrate test.

### Coagulase test

Coagulase is an enzyme that clots blood plasma. This test is carried out on Gram positive *Staphylococcus aureus*. A drop of sterile distilled water was placed on each end of a sterile slide. A colony of test organism was emulsified on each spot to make thick suspensions. A loopful of plasma was added to one of the suspension and mixed gently. The slide was examined for clumping or clotting of the organism within 10 seconds. Plasma was not added to the second suspension which serves as control.

### Urease test

This is used to identify those organisms that are capable of hydrolysing urea (bacteria that produce urease) to produce ammonia and carbon dioxide. It is primarily used to distinguish urease-positive protease from other Enterobacteriaceae. Organisms that hydrolyze urea rapidly (*Proteus* spp., *Morganella morganii*, and some *Providencia stuartii* strains) will produce strong positive reactions within 1 or 6 hours of incubation; delayed positive organisms (e.g. *Klebsiella* spp and *Enterobacter* species) will produce weak positive reactions in the slant in 6 hours of incubation which will be intense during further incubation. The culture medium will remain a yellowish color if the organism is urease negative e.g. *Escherichia coli*. If organism produces urease enzyme, the color of the slant changes from light orange to magenta. If organism do not produce urease the agar slant and butt remain light orange (medium retains original color).

## Results

Bacterial growths were observed after 24 hours of inoculation using the swab sticks, the observed growth are tabulated below.

**Table 1. Observed bacterial growth from inoculating swab sticks (mixed culture).**

| S/N | CODE ISOLATE | MEDIUM | GROWTH | COLOUR | SHAPE |
|---|---|---|---|---|---|
| 1 | 1A$^{ONE}$ | NA | + | CREAM | ROUND |
| 2 | 1A$^{TWO}$ | NA | + | CREAM | ROUND |
| | | | | ORANGE | ROUND |
| 3 | 1B$^{ONE}$ | NA | + | CREAM | ROUND |
| 4 | 1B$^{TWO}$ | NA | + | CREAM | ROUND |
| | | | | CREAM | FLAT |
| 5 | 2A$^{ONE}$ | NA | + | CREAM | ROUND |
| 6 | 2A$^{TWO}$ | NA | + | CREAM | ROUND |
| 7 | 2B$^{ONE}$ | NA | + | CREAM | ROUND |
| 8 | 2B$^{TWO}$ | NA | + | ORANGE | ROUND |
| | | | | CREAM | ROUND |
| 9 | 3A$^{ONE}$ | NA | + | CREAM | ROUND |
| 10 | 3A$^{TWO}$ | NA | + | CREAM | ROUND |
| 11 | 3B$^{ONE}$ | NA | + | ORANGE | FLAT |
| | | | | CREAM | ROUND |
| 12 | 3B$^{TWO}$ | NA | + | CREAM | SERATED |
| 13 | 4A$^{ONE}$ | NA | + | CREAM | ROUND |
| | | | | CREAM | ROUND |
| 14 | 4A$^{TWO}$ | NA | + | CREAM | ROUND |
| 15 | 4B$^{ONE}$ | NA | + | CREAM | SERATED |
| | | | | CREAM | ROUND |
| 16 | 4B$^{TWO}$ | NA | + | CREAM | ROUND |
| 17 | 5A$^{ONE}$ | NA | + | CREAM | FLAT |
| | | | | CREAM | ROUND |
| 18 | 5A$^{TWO}$ | NA | + | CREAM | SERATED |
| 19 | 5B$^{ONE}$ | NA | + | CREAM | ROUND |
| 20 | 5B$^{TWO}$ | NA | + | CREAM | SERATED |
| 21 | 6A$^{ONE}$ | NA | + | CREAM | ROUND |
| 22 | 6A$^{TWO}$ | NA | + | ORANGE | FLAT |
| 23 | 6B$^{ONE}$ | NA | + | CREAM | ROUND |
| | | | | ORANGE | ROUND |
| 24 | 6B$^{TWO}$ | NA | + | CREAM | ROUND |
| 25 | 7A$^{ONE}$ | NA | + | CREAM | ROUND |
| | | | | ORANGE | FLAT |
| 26 | 7A$^{TWO}$ | NA | + | CREAM | ROUND |
| 27 | 7B$^{ONE}$ | NA | + | CREAM | FLAT |
| | | | | CREAM | ROUND |
| 28 | 7B$^{TWO}$ | NA | + | ORANGE | ROUND |
| 29 | CONTROL | NA | - | - | - |

Table 1 shows the observed growth of the mixed culture that is the original isolates that were collected from the ATM machines themselves. There was growth on almost all the plates except controls, and the color on the plates were mainly cream and orange, while the shapes on the observed plates were mostly round, flat, and serrated.

Table 2 shows the second phase of culturing, where the original isolates were sub cultured into another set of plates. This time, it was observed that the colour of the colonies were mainly

**Table 2. Observation of mixed culture, sub-cultured onto another plate.**

| S/N | CODE ISOLATE | MEDIUM | GROWTH | COLOUR | SHAPE |
|---|---|---|---|---|---|
| 1 | 1A ONE | NA | + | CREAM | ROUND |
| 2 | 1A TWO | NA | + | CREAM | ROUND |
| 3 | 1A TWO | NA | + | ORANGE | ROUND |
| 4 | 1B ONE | NA | + | CREAM | SERATED |
| | | | | CREAM | ROUND |
| 5 | 1B TWO | NA | + | CREAM | FLAT |
| 6 | 1B TWO | NA | + | CREAM | ROUND |
| 7 | 1B TWO | | | ORANGE | ROUND |
| | | | | CREAM | SERATED |
| 8 | 2A ONE | NA | + | CREAM | ROUND |
| | | | | CREAM | SERATED |
| 9 | 2A TWO | NA | + | CREAM | ROUND |
| 10 | 2B ONE | NA | + | CREAM | ROUND |
| 11 | 2B TWO | NA | + | ORANGE | ROUND |
| 12 | 2B TWO | NA | + | CREAM | ROUND |
| 13 | 3A ONE | NA | + | CREAM | ROUND |
| 14 | 3A TWO | NA | + | CREAM | SERATED |
| | | | | CREAM | ROUND |
| 15 | 3B ONE | NA | + | ORANGE | ROUND |
| | | | | CREAM | SERATED |
| 16 | 3B ONE | NA | + | CREAM | ROUND |
| 17 | 3B TWO | NA | + | ORANGE | ROUND |
| | | | | CREAM | SERATED |
| 18 | 3B TWO | NA | + | CREAM | FLAT |
| 19 | 4A ONE | NA | + | CREAM | ROUND |
| | | | | CREAM | FLAT |
| 20 | 4A TWO | NA | + | CREAM | ROUND |
| 21 | 4B ONE | NA | + | CREAM | SERATED |
| 22 | 4B ONE | NA | + | CREAM | SERATED |
| | | | | CREAM | ROUND |
| 23 | 4B TWO | NA | + | CREAM | ROUND |
| 24 | 5A ONE | NA | + | CREAM | SERATED |
| | | | | CREAM | ROUND |
| 25 | 5A TWO | NA | + | CREAM | FLAT |
| 26 | 5B ONE | NA | + | CREAM | ROUND |
| 27 | 5B TWO | NA | + | CREAM | SERATED |
| 28 | 6A ONE | NA | + | CREAM | ROUND |
| 29 | 6A TWO | NA | + | ORANGE | ROUND |
| 30 | 6B ONE | NA | + | CREAM | ROUND |
| 31 | 6B ONE | NA | + | CREAM | SERATED |
| 32 | 6B TWO | NA | + | CREAM | ROUND |
| 33 | 7A ONE | NA | + | CREAM | SERATED |
| 34 | 7A ONE | NA | + | CREAM | ROUND |
| 35 | 7A TWO | NA | + | CREAM | SERATED |
| 36 | 7B ONE | NA | + | CREAM | ROUND |
| 37 | 7B ONE | NA | + | CREAM | SERATED |
| 38 | 7B TWO | NA | + | CREAM | ROUND |
| 39 | CONTROL | NA | - | - | - |

cream and few were orange, while there was almost an even distribution between the serrated and round shapes with few flat ones.

Table 3 shows further sub-culturing, this time of identified pure cultures with almost a repeat on what was observed in Table 2.

In summary, Tables 1–3 showed the bacteria isolated from the ATM machines at different times, with their shapes and color characteristics.

Table 4 shows the gram staining test for the isolates, with 9 (nine) isolates identified with almost even distribution between the gram positive and gram negative tests.

Table 5 showed the biochemical tests carried for identification of the microbes in the 9 isolates, with isolate 3 and 9 giving no microbial identity, while 7 distinct microbes were identified from plates 1, 2, 4, 5, 6, 7 and 8.

## Discussion

The study revealed that high levels of bacterial contamination were detected on electronic hardware user interfaces (ATM) examined in Federal Polytechnic Ede. ATMs harbours more bacterial contaminants and this can be attributed to their structural design and large surface area and this stems from the fact that ATMs are found in the open which actually encourages easy access from its users and as a result of that they are found in constant exposure to forces of nature namely wind and rain [7]. As at 2011, not much had been reported on bacterial contamination on ATMs or interfaces used in banks and cybercafés; the pioneering work of Oluduro et al., [28] has led to many studies confirming the relationship between bacterial contamination and ATM machines. The similarity in the bacterial loads recorded on interfaces studied can be attributed to frequent dermal contact and sharing by numerous users with differing hygiene practices and health conditions. The prevalence of microorganisms present on a surface is one of the microbe-associated factors that determine whether an infection will occur or not. The bacterial load on a fomite also determines the survival of bacteria on that fomite; the higher the concentration of a microorganism on a fomite the longer it survives and this invariably increases the chances of picking up the microbe from the environment [38, 39]. In addition, Neely and Maley [16] showed that microorganisms can survive for longer on plastics, the main material of which most accessible components of user interfaces are composed, than on other surfaces such as fabrics or steel. Thus, the ability of microorganisms to survive long on plastic user interfaces suggests the possibility of their serving as reservoirs for microorganisms and as a vehicle for their transfer. Rutala and Weber [40] had similar results with their bacterial counts on keyboards at a university health-care system. They found out that users' hands had more contamination than the surfaces themselves. The magnitude of the bacterial load on hands shows that users' hands are probably a major source of bacterial contamination on the interfaces, since on a daily basis hands typically touch a continuous sequence of surfaces, substances, objects, skin, food and body fluids. Bacterial contamination have also been found to be associated with currency notes, indicating the somewhat unhygienic rate of currency exchange between users [24, 41–44]. In spite of the amount of bacteria present at any one time, the type and quality of microorganism found on a surface is also an important determinant of whether an infection will occur or not. In this study, the automated teller machines examined were contaminated with considerable numbers of both Gram-positive and Gram-negative bacteria (Table 4); however, Gram-positive bacteria were found to occur more than Gram-negative bacteria (out of the nine isolates, five were Gram positive, while four were Gram negative). Most skin flora bacteria are Gram- positive, which would account for their predominance on the interfaces.

A total of seven (7) bacterial species were recovered from interfaces in this study, which included skin commensals (Staphylococcus and Micrococcus), environmental bacteria

**Table 3. Observed growth of pure culture of the isolates.**

| S/N | CODE ISOLATE | MEDIUM | GROWTH | COLOUR | SHAPE |
|-----|-------------|--------|--------|--------|-------|
| 1 | 1A$^{ONE}$ | NA | + | CREAM | ROUND |
| 2 | 1A$^{TWO}$ | NA | + | CREAM | ROUND |
| 3 | 1A$^{TWO}$ | NA | + | ORANGE | FLAT |
| 5 | 1B$^{ONE}$ | NA | + | CREAM | ROUND |
| 6 | 1B$^{ONE}$ | NA | + | CREAM | SERATED |
| 7 | 1B$^{TWO}$ | NA | + | CREAM | ROUND |
| 8 | 1B$^{TWO}$ | NA | + | CREAM | ROUND |
| 9 | 1B$^{TWO}$ | NA | + | CREAM | SERATED |
| 10 | 2A$^{ONE}$ | NA | + | CREAM | ROUND |
| 11 | 2A$^{ONE}$ | NA | + | CREAM | SERATED |
| 12 | 2A$^{TWO}$ | NA | + | CREAM | ROUND |
| 13 | 2B$^{ONE}$ | NA | + | CREAM | ROUND |
| 14 | 2B$^{TWO}$ | NA | + | ORANGE | ROUND |
| 15 | 2B$^{TWO}$ | NA | + | CREAM | ROUND |
| 16 | 3A$^{ONE}$ | NA | + | CREAM | ROUND |
| 17 | 3A$^{TWO}$ | NA | + | CREAM | ROUND |
| 18 | 3A$^{TWO}$ | NA | + | CREAM | SERATED |
| 19 | 3B$^{ONE}$ | NA | + | ORANGE | FLAT |
| 20 | 3B$^{ONE}$ | NA | + | CREAM | ROUND |
| 21 | 3B$^{TWO}$ | NA | + | CREAM | SERATED |
| 22 | 3B$^{TWO}$ | NA | + | CREAM | FLAT |
| 23 | 3B$^{TWO}$ | NA | + | ORANGE | ROUND |
| 24 | 4A$^{ONE}$ | NA | + | CREAM | ROUND |
| 25 | 4A$^{ONE}$ | NA | + | CREAM | ROUND |
| 26 | 4A$^{TWO}$ | NA | + | CREAM | ROUND |
| 27 | 4B$^{ONE}$ | NA | + | CREAM | SERATED |
| 28 | 4B$^{ONE}$ | NA | + | CREAM | ROUND |
| 29 | 4B$^{ONE}$ | NA | + | CREAM | SERATED |
| 30 | 4B$^{TWO}$ | NA | + | CREAM | ROUND |
| 31 | 5A$^{ONE}$ | NA | + | CREAM | FLAT |
| 32 | 5A$^{ONE}$ | NA | + | CREAM | ROUND |
| 33 | 5A$^{TWO}$ | NA | + | CREAM | SERATED |
| 34 | 5B$^{ONE}$ | NA | + | CREAM | ROUND |
| 35 | 5B$^{TWO}$ | NA | + | CREAM | SERATED |
| 36 | 6A$^{ONE}$ | NA | + | CREAM | ROUND |
| 37 | 6A$^{TWO}$ | NA | + | ORANGE | FLAT |
| 38 | 6B$^{ONE}$ | NA | + | CREAM | ROUND |
| 39 | 6B$^{ONE}$ | NA | + | ORANGE | ROUND |
| 40 | 6B$^{TWO}$ | NA | + | CREAM | ROUND |
| 41 | 7A$^{ONE}$ | NA | + | CREAM | ROUND |
| 42 | 7A$^{ONE}$ | NA | + | ORANGE | FLAT |
| 43 | 7A$^{TWO}$ | NA | + | CREAM | ROUND |
| 44 | 7B$^{ONE}$ | NA | + | CREAM | FLAT |
| 45 | 7B$^{ONE}$ | NA | + | CREAM | ROUND |
| 46 | 7B$^{TWO}$ | NA | + | ORANGE | ROUND |

(*Continued*)

**Table 3.** (Continued)

| S/N | CODE ISOLATE | MEDIUM | GROWTH | COLOUR | SHAPE |
|-----|--------------|--------|--------|--------|-------|
| 47 | CONTROL | NA | - | - | - |

*A^TWO = Sample collected in the afternoon from ATM dispensing ₦500.

*A^ONE = Sample collected in the morning from ATM dispensing₦500.

*B^ONE = Sample collected from ATM machine that dispenses ₦1000.

*B^TWO = Sample collected in the afternoon from ATM dispensing ₦1000.

*NA = Nutrient agar.

*+ = Growth on media.

*- = No growth on media.

(Serratia and Pseudomonas) and enteric bacteria (Salmonella and Klebsiella) (Table 5). Barbosa et al., [45] and Aquino et al., [46] in their study also discovered almost the same range of bacteria and this time it was found associated with multidrug resistance. The health risks associated with the majority of these bacteria are well documented [47]. The enteric bacteria encountered in this work are opportunistic human pathogens and have been associated with nosocomial infections [17, 48, 49]. The bacterial contaminants cultured from electronic hardware user interfaces are similar to bacteria that have been recovered from surfaces and objects in both hospital and non-hospital settings. Findings from other scientists have revealed similar results with mobile phones, environmental devices, [50, 51] currency notes [8, 24], day care centres [52], stethoscope covers[53] and computer keyboard and mouse interfaces [30, 54–56].

*Staphylococcus aureus*, was found to being the most frequent bacterial contaminant on the ATM machines. This result is similar to the report by Anderson and Palombo, [54] that *S. aureus* was the commonest isolate found to contaminate keyboards in a university setting. *Staphylococcus aureus* is a major component of the normal flora of the skin and nostrils, which probably explains its high prevalence as a contaminant, the prevalence of Staph lies on its easy dissemination resulting from human activities like sneezing, talking and contact with moist skin [31, 52, 57]. It's also been related with quite a number of infectious disease conditions and nosocomial infections [58]. Since users of such interfaces are human with human nature hence contamination of such surfaces with Staph is a common occurrence. Also, airborne organisms can be transported from users or passers-by. *Pseudomonas aeruginosa*, *Kliebsella spp*, *Bacillus spp*. and *S. epidermidis* were other major contaminants. The prevalence of these organisms on electronic hardware user interfaces is a cause for concern, because they have been shown to possess the potential to cause infections, especially in a hospital setting [48]. In different studies, each of these organisms has been implicated either as a major contaminant or as the most prevalent pathogenic bacteria recovered [40, 56, 59]. *Enterobacter aerogenes*, *Moraxella catarrhalis* and *Gaffkya tetragena* were the least frequent bacterial contaminants in most studies, although *Enterobacter aerogenes* and *Moraxella catarrhalis* have also been isolated from environmental objects. Fraser and Girling [56] recovered Moraxella spp. from

**Table 4. Gram staining of the isolates.**

| ORGANISM | 1 | 2 | 3 | 4 | 5 | 6 | 7 | 8 | 9 |
|----------|---|---|---|---|---|---|---|---|---|
| GRAM TEST | + | + | - | + | - | + | - | + | - |
| SHAPE | Slightly curved rod | Cocci | | Rods | Mucoid colony | Cocci in pairs | Straight rod | Rod | |

*+ positive,—negative.

**Table 5. Biochemical test for identification of the isolates.**

| ISOLATES | 1 | 2 | 3 | 4 | 5 | 6 | 7 | 8 | 9 |
|---|---|---|---|---|---|---|---|---|---|
| CATALASE | + | + | - | + | + | + | + | + | - |
| SUG. FERMENT. | - | - | - | + | - | - | + | + | - |
| CITRATE | + | - | - | - | + | - | + | + | - |
| INDOLE | - | - | - | + | - | - | - | - | - |
| COAGULASE | - | + | - | - | + | - | - | - | - |
| UREASE | - | - | - | - | - | - | - | - | - |
| OXIDASE | + | - | - | - | + | + | - | - | - |
| ORGANISM | Pseudomonas | Staphylococcus | | E. coli | Serratia | Micrococcus | Salmonella | Klebsellia | |

*+ = positive,— = negative.

keyboards in a veterinary practice. A high rate of contamination of user interfaces by bacteria was recorded in this study, as all interfaces sampled yielded bacterial isolates. This result was comparable to reported culture rates of over 70% in previous works [55, 56, 58, 59]. With such high levels of contamination found in association with people and the ease at which they are exposed in the environment, gives a cause for alarm [11, 60, 61]. Various bacterial species were found to coexist on an interface and on the hands of users. Interfaces harbour a community of bacteria with varying virulence and pathogenicity, thereby increasing the risk of infection and also the severity of infections [21, 62]. The conducive environment provided by interface users as a result of their unhygienic practices may account for this problem[16]. This could be related to the fact that multiple contaminations is influenced by the level of personal hygiene exhibited by users, since most display a poor level of hygienic practice during interface usage. Multiple contaminations was higher on keyboards and users' hands than on ATMs and mouse devices; the fact that keyboards are more frequently used than the other interfaces could explain the great diversity of bacteria found on them. Hands touch an array of different surfaces and objects, regularly picking up different types of bacteria from different fomites [9, 24, 41, 63, 64]. Although the ATM had the highest contaminant loads of bacteria [28, 31], the number of bacterial species was reduced; this can be attributed to the fact that users spend very little time on the ATM. Multiple contamination differs among different occupational groups and organizations; this could be attributed to differences in hygiene level among these occupational groups and organizational types. Fraser and Girling [56] showed that there exists a relationship between poor hygiene and high levels of bacterial contamination.

## Conclusion

This study has shown that the road is still long to achieve a satisfactory degree of social awareness especially towards infectious diseases. Indispensable ATM machines harbour a more significant and aggressive bacterial load than public toilets' flush buttons, an alarming finding that hopefully can modify individual behaviour while touching ATM keypads and touch screens. The selective preponderance of definite bacterial isolates on ATM key pads like gram-positive microorganisms and their paucity on the flush buttons of public toilets is a strange finding that needs further studies that will investigate the causes for such findings. In addition, further wider scale researches are required to isolate viral loads. The banks themselves should be held responsible for frequent cleaning of these devices as users are actually patronizing the banks and some increasing their profit margins. Anyway, the need for hand-sanitizers and hand wash points next to the ATM machines is an urgent necessity that should be implemented in the near future, an issue that deserves to be raised to the decision makers for better quality of life in our society.

## Recommendation

It is recommendable that public awareness organization or a recognized platform and the financial institutions in Nigeria such as the commercial banks, should create a grand slam awareness to kick against the unhygienic use of the automated teller machines as it is a public user interface that requires frequent use.

Users of the interfaces should in turn cultivate the use of sanitizers, in order to prevent themselves from bacterial microorganisms, contamination to the minimum. This measure is assured to pose no threat to any user of the automated teller machine.

Financial institutions can as well help provide sanitizers for the users of the machines and also put up the act of short period sanitations of the machine area at certain times of stipulated hours of the day.

## Supporting information

**S1 Table. Observed bacterial growth from inoculating swab sticks (mixed culture).**
(DOCX)

**S2 Table. Observation of mixed culture, sub-cultured onto another plate.**
(DOCX)

**S3 Table. Observed growth of pure culture of the iso.**
(DOCX)

**S4 Table. Gram staining of the isolates.**
(DOCX)

**S5 Table. Biochemical test for identification of the isolates.**
(DOCX)

## Author Contributions

**Conceptualization:** O. G. Dawodu.

**Data curation:** R. B. Akanbi.

**Investigation:** R. B. Akanbi.

**Methodology:** R. B. Akanbi.

**Project administration:** O. G. Dawodu.

**Supervision:** O. G. Dawodu.

**Writing – original draft:** R. B. Akanbi.

**Writing – review & editing:** O. G. Dawodu.

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
