## [Decision Letter · Decision Letter 0]

18 Feb 2021

PONE-D-21-02655

ISOLATION AND IDENTIFICATION OF MICROORGANISMS ASSOCIATED WITH AUTOMATED TELLER MACHINES ON FEDERAL POLYTECHNIC EDE CAMPUS

PLOS ONE

Dear Dr. Dawodu,

Thank you for submitting your manuscript to PLOS ONE. After careful consideration, we feel that it has merit but does not fully meet PLOS ONE’s publication criteria as it currently stands. Therefore, we invite you to submit a revised version of the manuscript that addresses the points raised during the review process.

We look forward to receiving your revised manuscript.

Kind regards,

Muhammad Sajid Hamid Akash

Academic Editor

PLOS ONE

Journal Requirements:

https://www.semanticscholar.org/paper/Microbial-Contamination-of-Currency-Notes-and-Coins-Alemu/917827b71571f500e8772265d8c63d81bb3a061d?p2df

https://www.academia.edu/3620737/ANTIBIOGRAM_OF_BACTERIA_ISOLATED_FROM_AUTOMATED_TELLER_MACHINES_WITHIN_ABAKALIKI_METROPOLIS

https://www.slideshare.net/NanaAfrahAntohene/dissertation1-62334948

https://media.neliti.com/media/publications/7173-EN-automated-teller-machine-atm-a-pathogen-city-a-surveillance-report-from-location.pdf

In your revision ensure you cite all your sources (including your own works), and quote or rephrase any duplicated text outside the methods section. Further consideration is dependent on these concerns being addressed.

3. In order to ensure your methods are reproducible, please state the name and location of the bank that was visited for sample collection in your manuscript.

"The funders had no role in study design, data collection and analysis, decision to publish, or preparation of the manuscript. "

At this time, please address the following queries

5. We note you have included a table to which you do not refer in the text of your manuscript. Please ensure that you refer to Tables 1-5 in your text; if accepted, production will need this reference to link the reader to the Table.

Reviewers' comments:

Reviewer's Responses to Questions

**Comments to the Author**

1. Is the manuscript technically sound, and do the data support the conclusions?

Reviewer #1: Yes

Reviewer #2: Partly

2. Has the statistical analysis been performed appropriately and rigorously? 

Reviewer #1: N/A

Reviewer #2: I Don't Know

3. Have the authors made all data underlying the findings in their manuscript fully available?

Reviewer #1: Yes

Reviewer #2: Yes

4. Is the manuscript presented in an intelligible fashion and written in standard English?

Reviewer #1: No

Reviewer #2: Yes

5. Review Comments to the Author

Reviewer #1: Manuscript Number: PONE-D-21-02655

Submission Title: ISOLATION AND IDENTIFICATION OF MICROORGANISMS ASSOCIATED WITH AUTOMATED TELLER MACHINES ON FEDERAL POLYTECHNIC EDE CAMPUS

Dear Authors,

Thanks for submitting the manuscript. No doubt, this research article is intriguing and informative. Your manuscript has been reviewed but it needs substantial revision. My point-to-point comments are listed below. You are encouraged to carefully revise the manuscript, highlighting the exact changes made.

1. According to my opinion, the introduction section needs more literature survey for better literature review.

2. Describe the need of the current study in the presence of other literature in the introduction as well as in conclusion.

3. There should be space between numerical value and its unit like 20 seconds. There is need to correct this mistake carefully in your manuscript.

4. According to my opinion, result section needs to be revised. First, the result should be clearly elaborated and defined. The table numbers should be mentioned in the text.

5. Sample Collection and Processing section demands revision as author has mentioned in abstract, “Swabs were collected from each ATM screen, buttons, floor, user’s hand, and exposure of plates” (Sample Collection and Processing does not provide any information regarding the places from where sample collect)

6. In my opinion, conclusion must be revised and author has mentioned recommendation in separate heading so there is need to remove recommendation word from conclusion heading.

7. Add the limitations of this paper.

8. Some English revision of the manuscript is needed.

Reviewer #2: There is no up to mark referencing. A lot of techniques used are without references. There is no proper method mentioned that how swabs having sample were transported from the venue of collection to working area of Laboratory. These swabs may be contaminated during transport process.

6. PLOS authors have the option to publish the peer review history of their article (what does this mean?). If published, this will include your full peer review and any attached files.

Reviewer #1: **Yes: **Yes , agree

Reviewer #2: No

---

## [Author Response · Author response to Decision Letter 0]

28 May 2021

Response to the Reviewer 1

The introduction and the conclusion sections have been updated with new info and the relevance of the study highlighted.

The space between SI units have been corrected throughout the manuscript.

The results section has been revised and explained and also mentioned in the text.

The place of collection of samples has been clearly stated.

The grammatical errors has been corrected.

The conclusion and the recommendation sections has been separated.

Reviewer 2

Dualization of references within the same paragraph has been corrected.

Repetitive sentences has been corrected

Clarity concerning transfer of swabs to the lab has been given.

The reference style has been updated to read the Vancouver format.

Abbreviations has been corrected and used properly.

The biochemical tests has been updated for the needed proper identification of the listed microbes.

The transport of the swabs to the lab has been clarified.

Editors Comments:

 The overlapping texts with subsequent literature has been corrected and references included in the paper.

The results section has been updated with explanatory tables.

The discussion has been reviewed to include the necessary tables in line with the text.

The journal has been corrected in line with PLOSOne specifications.

---

## [Decision Letter · Decision Letter 1]

1 Jul 2021

Isolation and identification of microorganisms associated with automated teller machines on Federal Polytechnic Ede campus

PONE-D-21-02655R1

Dear Dr. Dawodu,

We’re pleased to inform you that your manuscript has been judged scientifically suitable for publication and will be formally accepted for publication once it meets all outstanding technical requirements.

Kind regards,

Muhammad Sajid Hamid Akash

Academic Editor

PLOS ONE

Additional Editor Comments (optional):

Reviewers' comments:

Reviewer's Responses to Questions

**Comments to the Author**

1. If the authors have adequately addressed your comments raised in a previous round of review and you feel that this manuscript is now acceptable for publication, you may indicate that here to bypass the “Comments to the Author” section, enter your conflict of interest statement in the “Confidential to Editor” section, and submit your "Accept" recommendation.

Reviewer #1: All comments have been addressed

2. Is the manuscript technically sound, and do the data support the conclusions?

Reviewer #1: Yes

3. Has the statistical analysis been performed appropriately and rigorously? 

Reviewer #1: Yes

4. Have the authors made all data underlying the findings in their manuscript fully available?

Reviewer #1: Yes

5. Is the manuscript presented in an intelligible fashion and written in standard English?

Reviewer #1: Yes

6. Review Comments to the Author

Reviewer #1: (No Response)

7. PLOS authors have the option to publish the peer review history of their article (what does this mean?). If published, this will include your full peer review and any attached files.

Reviewer #1: **Yes: **Kanwal Irshad

---

## [Editor Report · Acceptance letter]

13 Jul 2021

PONE-D-21-02655R1 

Isolation and identification of microorganisms associated with automated teller machines on Federal Polytechnic Ede campus 

Dear Dr. Dawodu:

I'm pleased to inform you that your manuscript has been deemed suitable for publication in PLOS ONE. Congratulations! Your manuscript is now with our production department. 

Kind regards, 

on behalf of

Dr. Muhammad Sajid Hamid Akash 

Academic Editor

PLOS ONE